# Rapid Production of Carbon Nanotube Film for Bioelectronic Applications

**DOI:** 10.3390/nano13111749

**Published:** 2023-05-26

**Authors:** Hein Htet Aung, Zhiying Qi, Yue Niu, Yao Guo

**Affiliations:** School of Physics, Beijing Institute of Technology, Haidian, Beijing 100081, China; thusheinlin53@gmail.com (H.H.A.);

**Keywords:** carbon nanotube, flexible electronics, conductive film, electrocardiogram, bioelectronics

## Abstract

Flexible electronics have enormous potential for applications that are not achievable in standard electronics. In particular, important technological advances have been made in terms of their performance characteristics and potential range of applications, ranging from medical care, packaging, lighting and signage, consumer electronics, and alternative energy. In this study, we develop a novel method for fabricating flexible conductive carbon nanotube (CNT) films on various substrates. The fabricated conductive CNT films exhibited satisfactory conductivity, flexibility, and durability. The conductivity of the conductive CNT film was maintained at the same level of sheet resistance after bending cycles. The fabrication process is dry, solution-free, and convenient for mass production. Scanning electron microscopy revealed that CNTs were uniformly dispersed over the substrate. The prepared conductive CNT film was applied to collect an electrocardiogram (ECG) signal, which showed good performance compared to traditional electrodes. The conductive CNT film determined the long-term stability of the electrodes under bending or other mechanical stresses. The well-demonstrated fabrication process for flexible conductive CNT films has great potential in the field of bioelectronics.

## 1. Introduction

Flexible electronics have made significant advancements in the last two decades [1,2,3,4]. There are a wide variety of applications for flexible electronics, including strain sensors, flexible touch screens, transistors, energy harvesting and storage, and artificial skins [5,6,7,8]. It is reasonable to expect the large-scale commercialization of flexible electronics in the near future. As an important component of flexible electronic devices which have a huge impact on device performance. Currently, the most widely used conductive electrode is the indium tin oxide (ITO) transparent electrode [9]. Due to its cumbersome production process, high manufacturing cost, brittle texture, and poor flexibility, it cannot meet the development needs of flexible electronic devices [10]. Because of this, this research focuses on the design and preparation, performance regulation, and process exploration of the new flexible conductive film and conducts research on simple and low-cost preparation of high-performance flexible and conductive Caron nanotube (CNTs) film to replace traditional ITO electrodes. Established semiconductor manufacturing techniques are used for the majority of nano-electronic devices, and these devices are typically fabricated on solid, flat, and homogeneous surfaces [11]. The possibility of flexible, foldable, and embedded technologies has kindled interest in employing unusual materials such as soft plastic and nonplanar ones [12,13]. Despite this, flexible electronics still face several significant obstacles; one of the most significant is the achievement of conductive film that can sustain substantial elastomeric deformation while retaining an appropriate level of electrical conductivity. Flexibility is defined as the capacity to adapt to and recover from relative deformation [14,15], whereas excellent electrical performance is defined as the capacity to preserve electrical integrity when subjected to strain. Material science, nanotechnology, and electrical engineering have helped create flexible electronic devices. This has led to the development of nanomaterial synthesis, design, and manufacturing procedures that render flexible electronics functional, including nanosheets, nanotubes, and nanowires. Carbon nanotubes (CNTs) with inherent crystalline cylindrical structures, high Young’s modulus, suitable tensile strength, large elastic strain limit, large surface area, and the ability to form various macroscopic assemblies, are considered a perfect fit for flexible electronics [16,17,18,19,20,21,22,23,24,25]. Extensive research has been carried out to explore this promising nanoscale material, due to its potential technological and biomedical applications. Although individual CNTs possess excellent electronic and mechanical properties such as extremely high electron mobility (100,000 cm^2^/Vs) [26], high electrical conductivity (104 Scm^−1^) [27], high thermal conductivity (3500 W/mK) [28], and exceptionally good mechanical strength (tensile strength of 100 GPa for individual multi-walled tube) [29], the properties of the bulk material depend on how the CNTs are organized. While individual nanotubes are small and easy to manipulate, fabricating large-scale structures with precise control over the location of the CNTs’ building pieces is challenging. However, CNTs can be constructed into networks across enormous areas with features that are reminiscent of the individual tubes. For example, a field effect mobility of 1236 cm2/Vs [30], and electrical conductivity of about 12,825 Scm^−1^ have been reported for CNT networks [31]. Recent developments in filtering and purification technologies have made it possible to produce uniform CNT ensembles with different durations and electronic compositions [32,33]. Conductive CNTs membranes are one of the most promising and effective materials for flexible electronics. Researchers in the areas of physics and clinical medicine have shown tremendous interest in this issue for bioelectronics, for example, the monitoring systems of electrocardiograms (ECG) [34,35,36]. The gel-type silver/silver chloride (Ag/AgCl) electrode is the most widely used electrode for ECG measurements [37]. However, gelled electrodes cannot be used for long-term observation in devices [38]. These issues have been addressed using dry conductors [39]. Owing to their comfort and suitability for long-term monitoring, CNT-based flexible sensors have gained considerable interest [40,41,42]. Here, we developed a novel method to fabricate a CNT flexible electrode, which can enhance productivity with excellent uniformity, and used it as an ECG sensor patch, which showed comparable results to the traditional gel-type silver/silver chloride (Ag/AgCl) electrode. The novelty in this paper is the efficient method presented for fabricating flexible conductive CNT films on various substrates. The fabrication process is dry and solution-free, enabling mass production of the films. The CNT films exhibit satisfactory conductivity, flexibility, and durability. Moreover, we demonstrate the application of the CNT films in bioelectronics, by using them to collect electrocardiogram (ECG) signals with high fidelity and sensitivity, compared to traditional electrodes. The novelty of this work lies in the development of a rapid and cost-effective method for fabricating high-performance CNT films, which can be used in various bioelectronic devices. Furthermore, the use of these CNT films in ECG signal collection demonstrates their potential for a wide range of biomedical applications.

## 2. Results and Discussion

Raw CNTs were synthesized using chemical vapor deposition (CVD), as shown in Figure 1a and described in the Methods section. The visualization structure of the CNT consists of a cylindrical graphitic sheet rolled up with 100% black color. The average diameter and length of the CNTs are approximately 2–4 nm and 500 µm, respectively, and their surface area are ≥450 m^2^ g^−1^. Scanning Electron Microscopy (SEM) was used to observe micromorphology. The synthesized raw CNTs were bundled, as shown in Figure 1b. The bundling of the synthesized raw carbon nanotube causes agglomeration and nonuniformity, and the dispersion of the bundled sample is usually the first step towards a uniformly dispersed CNT film, which is a major challenge for the current mass production process, as discussed below.

Raman spectroscopy was conducted on the sample, as shown in Figure 1c. The Raman spectra of carbon nanotubes (CNTs) show several distinct features that can be used to study their properties. The major Raman peak in the spectra is the D-band, which appears at 1320 cm^−1^. The minor peaks at 1600 cm^−1^, which correspond to the G-band. There is a broad peak at 2650 cm^−1^, which is the G′ band. The D-band is a defect mode and its presence indicates that the sample has high quality defects. The G-band corresponds to the disorder-induced mode in CNTs, which results from the double resonance Raman scattering process [43]. The depth profile of the G-band of CNTs allowed us to choose the area of the sample that would yield the highest intensity of the Raman signal. In this case, the best signal was obtained at a depth of 1 μm in the sample. The study of the D-band and G′ band features has been important for studying trigonal warping effects in single-wall nanotubes (SWNTs) and for quantitatively measuring the effect of trigonal warping on the dispersion relations for electrons [44]. The G′ band is typically found at around 2650 cm^−1^ in the Raman spectrum of CNTs and is referred to as the second-order overtone of the G-band. The origin of the G′ band in CNTs is attributed to the trigonal warping effect, which arises due to the curvature of the nanotube structure. By analyzing the patterns and relationships between these bands, we can gain insights into the electronic and vibrational properties of CNTs.

The sample was further subjected to powder X-ray diffraction (XRD), as shown in Figure 1d. XRD was also used to study the mean diameter, diameter dispersity, and finite size of CNTs. It has been reported that the degree position of the peak and the width of the (1 0 0) peak are sensitive to the aforementioned parameters. The spectra of dispersive spectroscopy (EDS) and quantitative results of CNT arrays are presented in Figure 1e and Table 1, respectively. It also shows the EDS measurement results of the original CNTs. We found that the original element composition of CNT is nearly 100 wt% carbon atoms, and no other mixture was detected. The normalized EDS spectrum showed a single peak corresponding to the carbon K-alpha emission line, with an intensity proportional to the amount of carbon present in the sample as the sample is assumed to be 100% pure carbon. Transmission Electron Microscopy (TEM) morphology shows that CNT arrays are multi-walled nanotubes and that the diameter of the synthesized MWCNT was about ≥500 um.

In previous studies, the solution-based method was used to disperse bundled CNTs; however, it involves chemical dispersing agents that are difficult to remove and ultrasound that breaks the CNT [45,46,47]. To avoid the use of chemical dispersing agents and ultrasounds, we developed a novel method inspired by mechanical exfoliation, which has been widely used to disperse two-dimensional materials from the bulk, for example, monolayer or few-layer graphene from bulk graphite. The raw carbon nanotubes from the CVD synthesis were transferred onto one adhesive tape, and another adhesive tape was then placed onto the former one with the CNT sample. The attached tapes were pressed to ensure that the CNT sample had good contact with both tapes. The tapes were then separated, and then attached again with a displacement, and the CNTs were separated into regions. Such processes were repeated multiple times until the trace of the CNTs uniformly covered the entire target region of the tape. This method is comparable to the mechanical exfoliation of two-dimensional materials from bulk van der Waals materials, except that the exfoliated samples are CNT bundles, and the goal is to cover a uniform CNT/tape network rather than a thin layer of 2D materials. The method is easy to operate, as shown in Figure 2a and can be used in laboratories for research or in factories for mass production. This method is versatile and can be applied to different substrate films, such as commercial latex, scotch, and band-aid tapes, as shown in Figure 2b.

The properties of the fabricated CNT films were characterized. Figure 3a–d shows the SEM images of the CNT film. It can be seen clearly that the CNTs have been effectively dispersed, forming a uniform network on the tape. The SEM images taken from four different locations show that the CNTs were uniform across a large scale. We further conducted electrical measurements using the four-probe method, as shown in Figure 3e–h. The data were collected from the up, left right, and down sides of the CNT film, which show the sheet resistance of 12.35 Ω/sq, 56.62 Ω/sq, 43.37 Ω/sq, and 50.33 Ω/sq, respectively. The conductivity of the conductive CNT film fabricated using this method is comparable to that of the ITO film on a wafer scale, which is due to the high conductivity of the CNTs and their junctions, as demonstrated in previous studies [48,49]. Furthermore, the flexibility of the CNT film allows the deformation of the tape without worrying about the degeneration of the conductivity. As shown in Figure 3i–l, the conductivity of the conductive CNT film was maintained at the same level after 100 bending cycles. The sheet resistances are 9.06 Ω/sq, 56.62 Ω/sq, 45.37 Ω/sq, and 50.33 Ω/sq, respectively. Therefore, the conductive CNT film fabricated using a novel method has conductivity comparable to that of the ITO film and excellent flexibility.

We used a CNT conductive film as a flexible electrode for bioelectronics. The traditional ECG patch uses Ag/Cl gel as the electrode, which causes discomfort to the patient. In addition, the liquid in the gel-based electrodes evaporates, and the gel dries, which prevents long-term testing of the subject. The CNT conductive film was fabricated using a novel method and was used as the flexible ECG patch. The CNTs do not fully block the adhesiveness of the tape, which makes the tape sticky to the skin. As shown in Figure 4a, the ECG signals of a 30-year-old male subject were monitored under stable conditions using CNT/tape electrodes to evaluate the variability in the results for the usage of CNT/tape electrodes. Figure 4b shows that the ECG signal amplitude is steady. The ECG signals acquired using the CNT/tape electrodes were equivalent to those obtained using traditional gel electrodes, as shown in Figure 4c. This indicates that these CNT/tape electrodes are potential candidates for monitoring the ECG over the long term. The signal could be further fed into a machine learning model for automatic diagnosis, which provides great convenience to subjects that require portable and long-term ECG monitoring devices, as shown in Figure 4d.

## 3. Method

### 3.1. CNT Synthesis

Commercial CNTs were manufactured via chemical vapor deposition. This is achieved by using a carbon source in the gas phase and an energy source, such as a resistively heated coil, to impart energy to a gaseous carbon molecule. Commonly used carbon sources include methane, carbon monoxide, and acetylene. This results in the formation of CNTs. The CVD-based CNT synthesis is a two-step procedure. First, the metal catalyst is deposited on the surface of the reaction vessel; then, the catalyst is exposed to the carbon source gas. Secondly, the temperature for the synthesis of CNTs using this technique is generally 500–1000 °C [50].

### 3.2. Conductive Tape Fabrication

The homogeneous distribution of the CNT is crucial for obtaining a uniform conductive flexible substrate. The mechanical exfoliation method is a technique used to create ultra-thin two-dimensional materials such as graphene and other atomic thin van der Waals layers. Inspired by that, we use mechanical exfoliation as an intuitive method to disperse the CNTs on the flexible substrate. This method is cost-effective and can be applied to larger surface areas. Here’s a detailed step-by-step explanation of the process. The chosen tapes available are detachable, thin, and lightweight. Firstly, press the sticky side of the tape onto the bulk of synthesized CNT, and then carefully peel it off. This process will transfer the bulk of CNTs onto the tape. Then, another piece of tape is attached to the surface of the tape with CNT, followed by the peeling-off process; the bulk of CNT is thinned and dispersed for the first time. The attach-peeling process is repeated many times until the tape is uniformly covered with CNT, as shown in Figure 2a. The tape surface is examined to ensure that the CNTs are evenly spread out. After completing the above steps, we have a tape surface with a uniformly dispersed, ultralong CNT thin film that exhibits excellent conductivity. By following these steps, the mechanical exfoliation or tape peeling method can effectively create a thin film of carbon nanotubes on the surface of an adhesive tape, which can then be used for various applications requiring conductive surfaces.

### 3.3. Electrical Characterization

Current-voltage (I–V) curves were obtained using a Keithley 4200 Semiconductor Characterization System. Measuring I–V curves on several spots along the CNT makes it possible to check the uniformity of the electrical properties of the conductive film. The experimental setup consisted of source-measure units (SMUs), which were used to apply a voltage to the sample and measure the current flowing through it. The CNT/tape sample was mounted onto a probe station using a conductive adhesive. The probes were then connected to the SMUs. The sweeping voltage was applied, and the current was recorded at each voltage step. The I–V curves obtained from the SMU were analyzed to determine the electrical properties of the CNT/tape sample. The data were plotted on a graph of current versus voltage to obtain the I–V curve.

### 3.4. Morphology Characterization

The morphologies of the CNT arrays, CNT films, and graphitized CNT films were characterized by scanning Electron Microscopy (SEM) using a Carl Zeiss NTs GmbH SUPRA 55 cold field emission. The data obtained from the experiments were subjected to statistical analysis using appropriate software. The SEM was performed at an accelerating voltage of 20 kV. The SEM images were taken at a magnification of 10,000×.

### 3.5. Raman Spectroscopy, X-ray Diffraction, EDS Analysis and TEM Morphology

Raman spectra from CNTs were acquired using standard commercial micro-Raman spectrometers and lasers, using a back-scattering configuration for all measurements, and their excellent high-temperature stability and good thermal contact with the substrate. The raw CNTs powder diffraction was characterized using XRD. The CNT sample must be prepared by dispersing the CNTs in a solvent such as water, ethanol, or isopropyl alcohol. The CNTs must be well-dispersed and free from any impurities that could interfere with the XRD analysis. A high-intensity X-ray source, such as a copper radiation source, is used to irradiate the sample. The wavelength of the X-rays used in the analysis is typically 1.54 Å. The diffraction pattern produced by the CNTs is captured by a detector, which is typically a two-dimensional detector such as a CCD (Charge-Coupled Device) camera. The diffraction pattern produced by the CNTs is analyzed using software such as XRD analysis software (DIFFRAC.SUITE Software, Bruker D8 Advance X-ray powder diffractometer). The software is used to determine the position, intensity, and width of the diffraction peaks, which can be used to determine the crystal structure and size of the CNTs. The XRD data is interpreted to determine the crystal structure and size of the CNTs. The diffraction pattern produced by the CNTs is compared to the diffraction pattern of known crystal structures to determine the crystal structure of the CNTs. The purity of the produced materials in terms of their morphology and chemical composition was confirmed through the SEM images and the Energy Dispersive X-ray Spectroscopy spectra (Zeiss Supra55 scanning electron microscope by Zeiss, headquartered in Oberkochen, Germany). We have supported our results with TEM morphology to study the main construction of the type of CNTs.

### 3.6. Electrocardiogram Measurement

Electrocardiogram (ECG) measurement using CNT/tape electrodes is a non-invasive method for monitoring the electrical activity of the heart. Here are the technical details for conducting ECG measurements using CNT/tape electrodes by CNT. The resulting CNT/tape electrode is then cut to the desired size and shape. The CNT/tape electrodes are placed on the skin at specific locations to measure the electrical activity of the heart. The standard 12-lead ECG placement includes six limb leads and six chest leads. The electrical signals obtained from the CNT/tape electrodes are amplified and filtered using a signal amplifier and filter. The amplifier amplifies the electrical signals while the filter removes unwanted noise and interference. The amplified and filtered electrical signals are recorded using a computer or a dedicated ECG machine. The ECG recording shows the electrical activity of the heart over time, including the P wave, QRS complex, and T wave. The ECG recording is analyzed by a trained healthcare professional to detect abnormalities in the heart’s electrical activity. The analysis includes measuring the duration and amplitude of each wave, identifying any irregularities or arrhythmias, and comparing the results to standard ECG criteria. The ECG recording is then analyzed to detect any abnormalities in the heart’s electrical activity.

## 4. Conclusions

We designed and fabricated a new method of fabricating a highly flexible and conductive film of ultra-long carbon nanotubes on top of the commercial tape, which was demonstrated using the edge of commercial tape self-adhesion sticky glues between CNTs and the tape. Such high-performance CNT films possess several essential features, such as good alignment, long length, small diameter (few walls), high density, stretching, and good conductivity. Owing to their unique structure, excellent electronic performance, and remarkable mechanical properties, these films are promising for many advanced applications. ECG signals were also successfully achieved using the CNT/tape conductive patch. We have described the most straightforward method for depositing ultra-long carbon nanotube self-adhesive tape. In our mechanochemical approach, the target surface is activated by simple peeling of the adhesive tape. This is the first example of a mechanochemical process being peeled over target adhesive tape surfaces, which can be extended to other materials and systems. Some of the applications that we envision as practically relevant are tapes covered with other nanomaterials.

## Figures and Tables

**Figure 1 nanomaterials-13-01749-f001:**
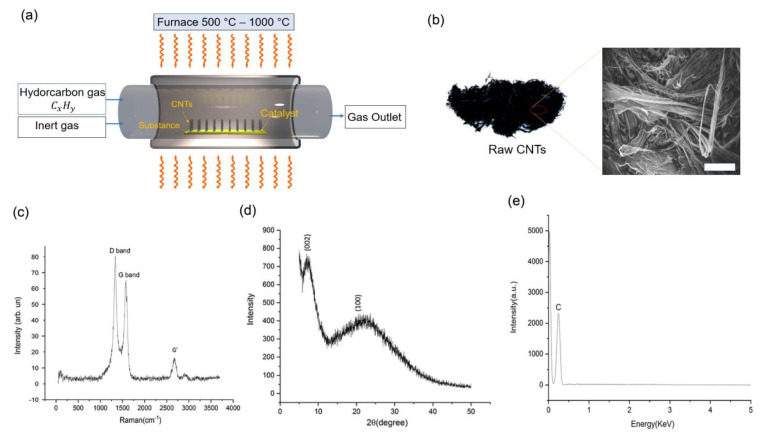
Raw CNT was synthesized (**a**) displays a graphic of the equipment used for CNT development by CVD in its simplest form. The technique requires passing a hydrocarbon vapor (usually for 15 to 60 min) through a tube furnace in which a catalyst material is present at sufficiently high temperature (500–1000 °C) to breakdown the hydrocarbon. When the system is cooled to room temperature, the CNTs that have grown over the catalyst may be collected. Heating a flask containing the liquid hydrocarbon (benzene, alcohol, etc.) causes it to evaporate, and the resulting vapor is then sucked into the reaction furnace through an inert gas. (**b**) Top-view scanning electron microscopic (SEM) images of continuously grown CNT films (scale bar 100 µm) with Morphological and electrical characterizations in various areas. (**c**) Raman spectra acquired at an excitation of 532 nm of the as-prepared carbon nanotube array. (**d**) Typical XRD analysis for CNTs. (**e**) Results of Energy dispersive spectroscopy (EDS) point analysis of CNT arrays.

**Figure 2 nanomaterials-13-01749-f002:**
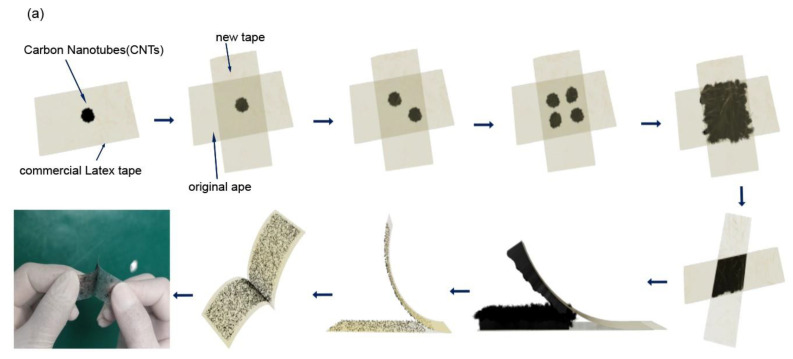
Fabrication and transfer process for CNT on tape (**a**) Schematics of the fabrication process for the conductive and flexible CNT images on tape substrate. (**b**) The image (1 cm × 1 cm size) developed method of CNT arrays was prepared using the modified parallel various tape-assisted, (**i**) CNT arrays on band-aid, (**ii**) CNT arrays on blue scotch tape, (**iii**) CNT arrays on white scotch tape, and (**iv**) CNT arrays on commercial latex tape.

**Figure 3 nanomaterials-13-01749-f003:**
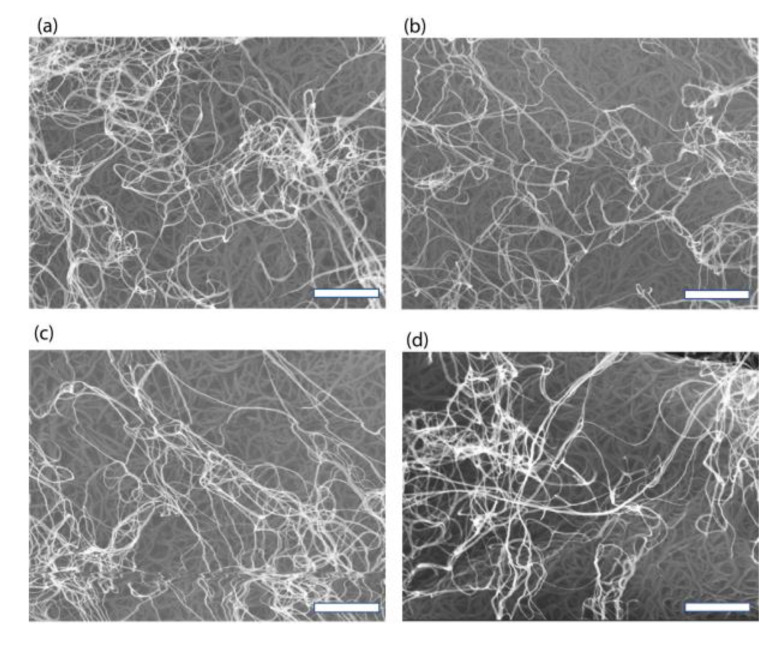
(**a**–**d**) the SEM image of the CNT film on several surfaces area (scale bar 1 µm) (**e**–**h**) the electrical measurement with the four-probe method. (**i**–**l**) the bending test of the conductivity of the conductive CNT film.

**Figure 4 nanomaterials-13-01749-f004:**
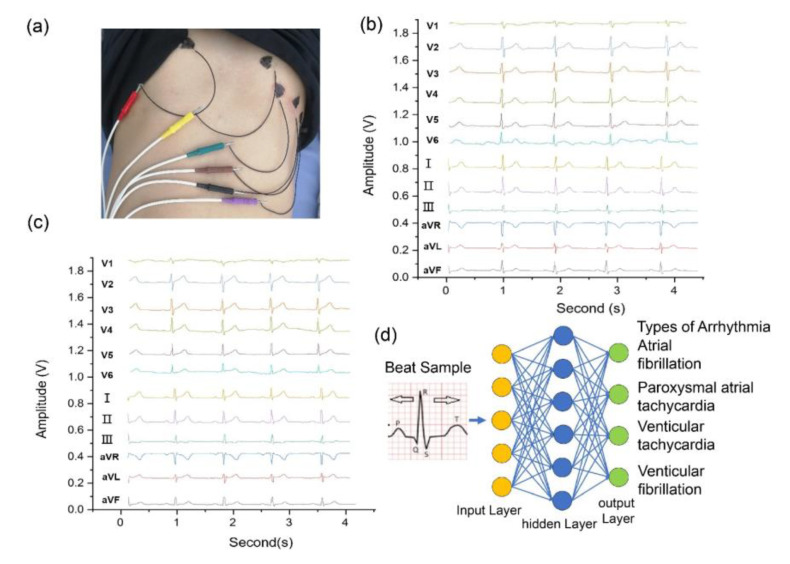
(**a**) shows a photograph of the flexible ECG patch. (**b**) ECG signal amplitude of CNT/tape electrodes. (**c**) ECG signals using the traditional electrode. (**d**) Segmentation of ECG images using an artificial neural network.

**Table 1 nanomaterials-13-01749-t001:** Energy dispersive spectroscopy (EDS) results from Figure 1e.

Point	Element	Weight%	Atomic%	Totals
Point 1	C K	100.00%	100.00%	100.00

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
