# Peer review of "Rapid Production of Carbon Nanotube Film for Bioelectronic Applications"

_nanomaterials, 2023, doi:10.3390/nano13111749_

Round 1

Reviewer 1 Report (Previous Reviewer 2)

I have already evaluated the manuscript during the previous submission and I think that now the authors have almost completely taken my comments into consideration.

So now , in my opinion, the manuscript is improved and it could be published

the english is ok

Author Response

Reviewer 2 Report (Previous Reviewer 1)

The paper is devoted to the development of a new method for creating electrodes from CNT grown by the well-known CVD method. The method of creating electrodes consists in sequential exfoliation on medical tapes. The resulting CNTs were studied by Raman and electrical measurements. Unfortunately, there are a number of comments to the work:

  1. Line 92 onwards. The authors first introduce the G-band into the discussion, and then, much further down the text, indicate its origin. This raises questions when reading. I recommend that you first describe all the Raman bands, and then discuss the patterns that connect them.
  2. Line 97-98. It is not clear why the authors believe that the concentration of defects in CNT is low (what exactly?). The presence of the G-band is given as an argument, which raises questions.
  3. Line 99. The nature of the Raman G' band is also remains unclear.
  4. It is not clear by which microscopic method the images in fig. 2(b). In addition, the figure does not show the scale.
  5. It is not very clear why the authors present 4 similar CNT images in Figures 3 (a-d). It is clear that the authors used more than 4 figures for the analysis. In my opinion, it is enough to cite one with a note that it is typical.
  6. It is not clear how the measurement results presented in Figures 3(e-h) and 3(i-l) differ. These 4 different measurements correspond to the 4 SEM images in Fig. 3(a-d)?
  7. It is not clear what caused the spread (by almost 1 order of magnitude) of the measured resistance value of different samples.
  8. Have these electrodes been used for long-term measurements?
  9. At this stage of the work, the advantages of the new electrodes are not clear. There is only the potential applicability of the new electrodes in long-term measurements declared by the authors.
  10. Also, the stability of the obtained electrodes to repeated bending and the effects of biological factors, such as sweating by the patient, remained unclear.

The presence of these comments indicates a certain incompleteness of the paper, both in terms of presenting the data and in terms of the application. I would recommend the authors to bring the applied part of the work to a more complete form.

Author Response

Reviewer 3 Report (New Reviewer)

1-     The main gain results must be inserted in the abstract

2-     The novelty in the paper is not clear, and the different in this work in compared to others must be inserted in the introduction part

3-     The XRD card number must be inserted

4-      It was recommended to support your results with TEM to study the main construction of the type of tube

5-     Also, the purity of the CNTs from any impurities was recommended to be investigated by EDX

It is accpetable

Round 2

Reviewer 2 Report (Previous Reviewer 1)

The authors eliminated most of the essential comments on the work. There remains the question of approbation of the practical application of these electrodes, however, I think that this can be safely taken out of the scope of this paper. I think that the paper can be published.

Reviewer 3 Report (New Reviewer)

The paper can be published in the current form 

This manuscript is a resubmission of an earlier submission. The following is a list of the peer review reports and author responses from that submission.

Round 1

Reviewer 1 Report

The paper is devoted to the discussion of the creation of electrodes for ECG using commercial CNT and commercial tape. In my opinion, the results are of undoubted engineering significance and can be applied in practice. However, I do not see a significant scientific component in the work. The authors use well known objects (CNT) obtained by the standard method (CVD). In this regard, I see no reason to publish this work in the journal Nanomaterials. I would recommend authors to apply to a journal with a more technical focus.

In addition, there are a number of comments on the text of the work:

  1. In my opinion, the term "Intuitive" is not entirely scientific. To be honest, when reading the title, it remains unclear what the authors wanted to say using this term ... Perhaps it would be worth replacing it, for example, with: low-cost, novel?
  2. Line 24. Repeat "artificial skins" twice in one sentence. It's probably a typo.
  3. Line 53. The term «relative» implies a comparison with some reference value. What reference value do the authors have in mind in this case?
  4. Line 58-60. At first glance, the authors contradict themselves, first mentioning that temperatures of 500-1000C are typical in the CVD method, and then stating that this method makes it possible to obtain CNT at room temperature. This place needs to be clarified.
  5. Line 65. The sentence "To achieve a uniform dispersion of the CNTs on commercial tape" is not grammatically consistent - the verb is missing. What did the authors want to say?
  6. Line 86-92. In my opinion, there is an unnecessary repetition of the description of the receipt of CNT.
  7. The authors begin to operate with bands in the Raman spectrum without explaining to the reader what these bands are and what causes them. The authors give an explanation later, but it is better to give it first.
  8. The sentence "The Raman spectra show that the major peaks are at 1250 cm-1 , which corresponds to the modes." dont clear. What modes do the authors have in mind?
  9. Line 147. What kind of previous research are you referring to?

In general, the quality of the text of the paper is low, contains many inaccuracies and typos. The clarity and scientific style also leave much to be desired. I recommend the authors to significantly revise the text for further submission of material to other journals.

Reviewer 2 Report

In this paper the authors present an interesting and simple approach for the fabrication of conductive CNT based electrodes and substrates. They discuss their use in flexible electronics components. The fabricated films are presented to be conductive, flexible and durable. The fabrication process is based on a simple tape exfoliation and is then applied to collect, as test application, ECG signal with good results.

the manuscript is for sure interesting, but in the current version it cannot be accepted for publication.

the discussion on the experimental methods, the introduction. and the discussion of the results sound really too simple and limited. no details are reported, and this is really bad considering that this wants to be a scientific paper. 

I recommend the authors a radical revision and to pay major attention to the discussion on how they performed all the steps in the fabrication and characterization

I also recommend a better introduction mentioning the different potential application of CNT in flexible devices (see for example: 

-Applied Physics Reviews 8 (4), 041325

-Chemosensors 202210(6), 223; https://doi.org/10.3390/chemosensors10060223

-Electrochimica Acta Volume 415, 20 May 2022, 140239

 -https://doi.org/10.1002/adfm.202110417

-DOI: 10.1109/JSEN.2022.3198847

)

along the text several english grammar errors can be find. I recommend a deep check and revision

Round 2

Reviewer 1 Report

The authors responded to all minor comments on the text, which made the text of the work much better. The authors also significantly improved the introduction, highlighting in detail the development and application of flexible electronics and sensors, which also significantly affected the quality of the text.

The key to the work of the authors is the creation of an array of CNT on commercial tape by exfoliation. The authors use commercial CNTs made by the well known CVD method. The exfoliation used by the authors has also been known for a long time. Nevertheless, the results obtained by the authors are of high practical importance in medicine and other areas. I think that the work can be published.

Reviewer 2 Report

I report here my previous comments:

>>The manuscript is for sure interesting, but in the current version it cannot be accepted for publication. the discussion on the experimental methods, the introduction. and the discussion of the results sound really too simple and limited. no details are reported, and this is really bad considering that this wants to be a scientific paper.

>>I recommend the authors a radical revision and to pay major attention to the discussion on how they performed all the steps in the fabrication and characterization. I also recommend a better introduction mentioning the different potential application of CNT in flexible devices (see for example: -Applied Physics Reviews 8 (4), 041325 -Chemosensors 2022, 10(6), 223; https://doi.org/10.3390/chemosensors10060223 -Electrochimica Acta Volume 415, 20 May 2022, 140239 -https://doi.org/10.1002/adfm.202110417 -DOI: 10.1109/JSEN.2022.3198847) 

The updated version of the manuscript submitted by the authors reports almost zero improvements in the methods section. As reported before, this cannot be considered a scientific paper if the technical methods are not described in details

I cannot recommend the pubblication 

the english is rather ok